# Synthesis and Properties of 1,3-Disubstituted Ureas Containing (Adamantan-1-yl)(phenyl)methyl Fragment Based on One-Pot Direct Adamantane Moiety Inclusion

**DOI:** 10.3390/molecules28083577

**Published:** 2023-04-19

**Authors:** Vladimir D’yachenko, Dmitry Danilov, Yaroslav Kuznetsov, Semyon Moiseev, Vladimir Mokhov, Vladimir Burmistrov, Gennady Butov

**Affiliations:** 1Department of Technology of Organic and Petrochemical Synthesis, Volgograd State Technical University (VSTU), 28 Lenin Avenue, Volgograd 400005, Russiagmbutov@mail.ru (G.B.); 2Volzhsky Polytechnic Institute (Branch), Volgograd State Technical University (VSTU), 42a Engels Street, Volzhsky 404121, Russia

**Keywords:** adamantane, isocyanate, urea, 1,3-disubstituted urea, soluble epoxide hydrolase, hsEH

## Abstract

A one-stage method for the preparation of 1-[isocyanato(phenyl)methyl]adamantane containing a phenylmethylene fragment located between the adamantane fragment and the isocyanate group, and 1-[isocyanato(phenyl)methyl]-3,5-dimethyladamantane with additional methyl groups at the nodal positions of adamantane, with a yield of 95% and 89%, respectively, is described. The method includes the direct inclusion of an adamantane moiety through the reaction of phenylacetic acid ethyl ester with 1,3-dehydroadamantane or 3,5-dimethyl-1,3-dehydroadamantane followed by the hydrolysis of the obtained esters. The reaction of 1-[isocyanato(phenyl)methyl]adamantane with fluorine(chlorine)-containing anilines gave a series of 1,3-disubstituted ureas with 25–85% yield. 1-[Isocyanato(phenyl)methyl]-3,5-dimethyladamantane was involved in the reactions with fluorine(chlorine)-containing anilines and trans-4-amino-(cyclohexyloxy)benzoic acid to obtain another series of ureas with a yield of 29–74%. The resulting 1,3-disubstituted ureas are promising inhibitors of the human soluble epoxide hydrolase (hsEH).

## 1. Introduction

Lipophilic fragments of inhibitors of soluble epoxidhydrolase (sEH, E. C. 3.3.2.10), an enzyme located in the arachidonic cascade [1,2,3,4] and involved in the metabolism of epoxy fatty acids (arachidonic acid metabolites) to the corresponding vicinal diols by catalytic addition of water molecules, usually contain adamantane [2] or aromatic fragments [5] in their structure. Inhibition sEH allows for successfully fighting against kidney diseases [6], cardiovascular diseases, and diabetes [7]. However, there are no references in the literature on sEH inhibitors, the lipophilic part of which contains both adamantane and an aromatic moiety at the same time. Vazquez et al. considered the replacement of the adamantane fragment with compounds containing polycyclic hydrocarbons smaller or larger than adamantane. Inhibitory activity values (IC_50_) for these compounds were 0.4–21.7 nM, indicating that sEH is able to accommodate inhibitors of very different sizes. However, it has been noted that the human liver microsomal stability of diamantane-containing inhibitors is lower than that of their corresponding adamantane counterparts [8].

The presence of closely spaced, bulk fragments of adamantyl (3,5-dimethyladamantyl) and phenyl in the structure of 1,3-disubstituted urea molecules will allow us to clarify the limiting dimensions of structures that can be used as sEH inhibitors, since the catalytical center of the enzyme is a “tunnel” of limited dimensions, with the following geometrical parameters: d(O2–H2) = 0.89 (2) Å, d(H2···O1’) = 1.78(2) Å, d(O2···O1’) = 2.6631(11) Å, (O2-H2···O1’) = 171 (2)°, symmetry operation 1–x, 2–y, 1–z [8]. Due to various intermolecular interactions [9], including nonclassical hydrogen bonds, centrosymmetric dimers formed by the classical hydrogen bond O2–H2···O1’ form a three-dimensional crystal structure with a rather high packing density of 71.1% [10]. 

One of our studies was devoted to the synthesis of symmetric 1,3-disubstituted diureas containing both an adamantane fragment and an aromatic ring in the lipophilic part [10]. Compounds containing bulky lipophilic fragments can be used to study features of inhibition between soluble epoxide hydrolases of different species due to the differences in the protein structures [11]. However, compounds containing in the right part the halogen-containing (F, Cl) anilines have not been previously obtained.

Thus, the synthesis of new compounds containing (adamantane-1-yl)(phenyl)methyl or (3,5-dimethyladamantane-1-yl)(phenyl)methyl fragments, and the synthesis of 1,3-disubstituted ureas based on them, is of significant scientific and practical interest.

## 2. Results and Discussion

1,3-Dehydroadamantane **2a** and 1,3-dehydro-5,7-dimethyladamantane **2b** were obtained by the well-known method [12]. Then they were involved in the reaction with phenylacetic acid ethyl ester. The adamantylation reaction of phenylacetic acid ethyl ester proceeded into the α-position to the carbonyl group, with the obtaining of ethyl esters of (±)-(adamantane-1-yl)phenylacetic acid **3a** and (±)-3,5-dimethyl-(adamantane-1-yl)- phenylacetic acid **3b** acids, yielding 91% and 85%, respectively (Figure 1) [13]. 

Obtained adamantyl containing derivatives of phenylacetic acid ethyl ester **3a** and **3b** were hydrolyzed in ethylene glycol in the presence of KOH at the temperature of 190 °C [10], to produce (±)-(adamantane-1-yl)phenylacetic acid **4a** and (±)-3,5-dimethyl- (adamantane-1-yl)phenylacetic acid **4b**, with yields of 67% and 73%, respectively.

Using a one-stage method which excludes the use of toxic and explosive reagents [10], acting on acids **4a** and **4b** with equimolar amounts of diphenylphosphoryl azide (DPPA) and triethylamine in toluene medium led to (±)-1-[isocyanato(phenyl)methyl]adamantane **5a** and (±)-1-[isocyanato(phenyl)methyl] -3,5-dimethyladamantane **5b**, with yields of 95% and 89%, respectively. (±)-1-[Adamantyl(phenyl)methyl]amine hydrochloride **6a** was obtained under mild conditions in toluene at room temperature using concentrated hydrochloric acid [14] with 60% yield (Figure 2).

Based on the obtained (±)-1-[isocyanato(phenyl)methyl]-3,5-dimethyladamantane **5b,** the respective urea compounds containing the (3,5-dimethyladamantane-1-yl)(phenyl)methyl fragment were synthesized by two methods (methods A and B).

For the synthesis of 1,3-disubstituted urea **8a**–**k** from isocyanates **5a** and **5b** according to method A, we have chosen halogen-containing (F, Cl) anilines **7a**–**e**, and trans-4-amino-(cyclohexyloxy)benzoic acid **7f**, on the basis of which the most active inhibitors of soluble epoxide hydrolase (sEH) were previously obtained [15] (Figure 3, Table 1).

1,3-disubstituted ureas **8a**, **8b** and **10a**–**c** were synthesized by method B from (±)-1-[adamantyl(phenyl)methyl]amine hydrochloride **6a** and aromatic isocyanates **9a**–**d**, as well as cyclohexyl isocyanate **9e**, as the closest analog trans-4-amino-(cyclohexyloxy)benzoic acid **7a** devoid of an oxophenylcarboxylic fragment (Figure 4).

Synthesis of 1,3-disubstituted urea **8a**–**k** and **10a**–**c** was carried out in an anhydrous diethyl ether medium for 12 h at room temperature in the presence of an equimolar amount of triethylamine (Table 1). Diethyl ether and triethylamine were chosen as the solvent and the base, respectively, for this reaction for a number of reasons. Ureas are usually insoluble in ether while most of the amines and isocyanates as well as triethylamine are soluble. For the cases when the starting material is insoluble in ether, DMF is used. As for the water and alcohols, they cannot be used as solvents for this reaction because they will react with the isocyanates. Most of the other polar solvents such as ethyl acetate can dissolve the resulting ureas which makes isolation more difficult. Inorganic bases are also insoluble in most of the solvents suitable for this reaction.

In ^1^H NMR spectra of compounds **8a**–**e** obtained from (±)-1-[isocyanato(phenyl)methyl]adamantane **5a**, the chemical shift of protons ^1^NH is within the range of 6.64–6.92 ppm, and the proton signals ^3^NH bound to anilines shift to a weaker field of 8.40–8.71 ppm, which is probably due to the close location of the electron-withdrawing phenyl substituent to the NH-group. In ^1^H NMR spectra of compounds **8f**–**j** obtained from (±)-1-[isocyanato(phenyl)methyl]-3,5-dimethyladamantane **5b**, the signals of ^1^NH proton shift to a strong field of 4.05 ppm compared to compounds **8a**–**e**, which is probably due to the presence of electron-donating methyl substituents in the nodal positions of adamantane. The proton signals ^3^NH bound to the phenyl substituent stay at the same range of 8.36–8.59 ppm, as for the compounds **8a**–**e**. For the compound **10c** obtained from (±)-1-[isocyanato(phenyl)methyl]adamantane **5a** and cyclohexyl isocyanate **9e**, in the absence of a phenyl substituent, the proton signal ^3^NH shifts to a strong field of 6.37 ppm. Similarly to compounds **8a**–**j**, the signal of a proton ^1^NH shifts to a strong field of 5.73 ppm.

The calculated lipophilicity coefficient LogP for the series of ureas **8a**–**k** is in the range of 5.96–6.93, which somewhat exceeds the allowable limits according to the Lipinski rule [17]. For a series of ureas **8f**–**j** obtained from (±)-1-[isocyanato(phenyl)methyl]-3,5-dimethyladamantane **5b**, the lipophilicity coefficient is 0.12 units higher than that of ureas **8a**–**e** obtained from (±)-1-[isocyanato(phenyl)methyl]adamantane **5a** (Table 1). Based on the literature data [11], such compounds will have a higher inhibitory activity, but have a lower solubility in water and are more susceptible to metabolism in vivo. Comparing urea **8k** obtained from *trans*-4-amino-(cyclohexyloxy)benzoic acid with analogs, it can be seen that the lipophilicity coefficient also became 0.12 units higher than for its analog **11** not containing methylene substituents in the nodal positions of adamantane. Comparing urea **8k** with previously obtained analogs **12**, **13** containing a fragment of *trans*-4-amino-(cyclohexyloxy)benzoic acid, it can be seen that the introduction of substituents in the nodal positions or in the bridge separating the adamantane fragment and the ureide group leads to an increase in the lipophilicity coefficient by 1.70 units (Table 1). 

The introduction of methyl substituents in the nodal positions of adamantane made it possible to reduce the melting temperatures of the ureas **8f**–**j** (51–196 °C) obtained from (±)-1-[isocyanato(phenyl)methyl]-3,5-dimethyladamantane **5b** by 37–198 °C, in comparison with the melting temperatures of similar urea products **8a**–**e** (233–270 °C) derived from (±)-1-[isocyanato(phenyl)methyl]adamantane **5a**. The general rule that lowering melting point increases solubility is based on the simplified solubility equation proposed by Wouters and Quéré [18]. Reduced melting point is also a positive factor for drug candidates as it simplifies preparation of drug dosage forms by hot-melt extrusion [19]. For urea **8k** obtained from isocyanate **5b** and *trans*-4-amino-(cyclohexyloxy)benzoic acid, the addition of methylene substituents to the nodal positions of adamantane leads to an increase in the melting temperature by 11 °C. Thus, the melting temperature of urea **8k** is 174 °C, and for urea **11** obtained from isocyanate **5a**, it is 163 °C. An increase in the melting temperatures by 58 °C is also observed in urea products obtained from 1-isocyanatomethyl-3,5-dimethyladamantane **12** and 1-isocyanatomethyladamantane **13** (Table 1). However, when comparing urea **8k** obtained from isocyanate **5b** with urea **12** obtained from 1-isocyanatomethyl-3,5-dimethyladamantane, the melting point decreases by 66 °C when introducing a phenyl substituent into the structure of isocyanate. A similar decline of melting temperatures by 19 °C is observed in urea **13** obtained from 1-[isocyanato(phenyl)methyl]adamantane **11** and 1-isocyanatomethyladamantane **8k** (Table 1, Figure 1).

## 3. Materials and Methods

### 3.1. Chemistry

Triethylamine (BioUltra ≥ 99.5%, CAS 121-44-8), 3-chloroaniline (99%, CAS 108-42-9), cyclohexyl isocyanate (98%, CAS 3173-53-3), phenyl isocyanate (98%, CAS 103-71-9) manufactured by Sigma-Aldrich (St. Louis, MO, USA) were used without purifying.

4-(Trifluoromethoxy) isocyanate (97%, CAS 35037-73-1), aniline (99+%, CAS 62-53-3), 2-fluoroaniline (99%, CAS 348-54-9), 3-fluoroaniline (98%, CAS 372-19-0), 4-fluoroaniline (99%, CAS 371-40-4) produced by the AlfaAesar (Ward Hill, MA, USA) were used without additional purification.

Diethyl ether was purified by well-known methods. *trans*-4-Amino-(cyclohexyloxy)benzoic acid **7a** [2], 1,3-dehydroadamantane **2a** [12], 1,3-dehydro-5,7-dimethyladamantane **2b** [12], 2-(adamantane-1-yl)-2-phenylacetic acid ethyl ester **3a** [15], 2-(adamantane-1-yl)-2-phenylacetic acid **4a**, 1-(isocyanato(phenyl) methyl)adamantane **5a** [10] were obtained by well-known methods.

### 3.2. Equipment

Purification of the obtained adamantyl-containing derivatives of phenylacetic acid ethyl ester **3a** and **3b** was performed on a Pure C-815 Flash Advanced chromatographic system (Buchi Labortechnik AG, Flawil, Switzerland).

Hydrolysis of the obtained adamantyl-containing derivatives of phenylacetic acid ethyl ester **3a** and **3b** was carried out on a Monowave 450 microwave laboratory reactor (Anton Paar GmbH, Graz, Austria).

The structure of the obtained compounds was confirmed by ^1^H, ^13^C, and ^19^F NMR spectroscopy, chromatography-mass spectrometry, and elemental analysis. Mass spectra were recorded on an Agilent GC 7820A/MSD 5975 chromatography-mass spectrometer (Agilent Technologies, Santa Clara, CA, USA) in fullscan (EI) mode. ^1^H NMR was performed on Bruker DPX 300 (Bruker Corporation, Billerica, MA, USA) in DMSO-*d_6_* solvent; chemical shifts ^1^H are given relative to SiMe_4_. Elemental analysis was performed on a Perkin-Elmer Series II 2400 instrument (Perkin-Elmer, Waltham, MA, USA). Melting points were determined using an OptiMelt MPA100 instrument (Stanford Research Systems, Sunnyvale, CA, USA).

### 3.3. Synthesis

(±)-2-(3,5-Dimethyladamantane-1-yl)-2-phenylacetic acid ethyl ester (**3b**). 

An amount of 2.5 g (0.015 mol) of 1,3-dehydro-5,7-dimethyladamantane **2b** was added to 10 g (0.061 mol) of phenylacetic acid ethyl ester. The reaction mixture was exposed at the temperature of 55–60 °C for 4 h. The excess amount of phenylacetic acid ethyl ester was distilled, and the resulting product was recrystallized from diethyl ether. The yield is 4.3 g (85%), colorless viscous liquid. Mass spectrum, m/z (Irel. %): 326 (15% [M]^+^), 253 (56% [(CH_3_)_2_-Ad-CH-Ph]^+^), 163 (100% [(CH_3_)_2_-Ad]^+^) (Appendix A). Anal. Calc. for C_18_H_22_O_2_: C 80.94; H 9.26. Found: C 80.92; H 9.28. M = 326.22.

(±)-2-(3,5-Dimethyladamantane-1-yl)-2-phenylacetic acid (**4b**). An amount of 4.0 g (0.012 mol) of 2-(3,5-dimethyladamantane-1-yl)phenylacetic acid ethyl ester (**3b**) was added to 7.0 g (0.125 mol) KOH in 70 mL of ethylene glycol. The reaction mixture was exposed at 190 °C for 16 h. The cooled reaction mass was diluted with 100 mL of H_2_O and extracted with ethyl acetate. The aqueous layer was placed in a rotary evaporator to remove ethyl acetate residues and concentrated hydrochloric acid was added to it until pH = 3. The precipitated white residual matter was filtered and dried in vacuum. The yield is 3.6 g (73%), white powder, m.p. 251–252 °C. Mass spectrum, m/z (I_rel._ %): 298 (14% [M]^+^), 163 (2% [(CH_3_)_2_-Ad]^+^), 135 (100% [(Ph)-CH-COOH]^+^) (Appendix A). Calc. for C_20_H_26_O_2_: C 80.50; H 8.78. Found: C 80.48; H 8.80. M = 298.19.

(±)-1-(Isocyanato(phenyl)methyl)-3,5-dimethyladamantane (**5b**). A mixture of 3.5 g (11.7 mmol) (3,5-dimethyladamantane-1-yl)phenylacetic acid **4b** and 2.37 g (23.4 mmol) triethylamine in 40 mL of anhydrous toluene was treated dropwise with 3.22 g (11.7 mmol) diphenylphosphorylazide at room temperature for 30 min. Then the reaction mixture was heated to boil and exposed for 30 min until the nitrogen release was completely stopped. The toluene was evaporated, and the product was extracted from the reaction mass with anhydrous diethyl ether. The yield is 3.1 g (89%), colorless crystals, m.p. 110–111 °C. Mass spectrum, m/z (I_rel._ %): 295 (3% [M]^+^), 163 (100% [(CH_3_)_2_-Ad]^+^) (Appendix A). Calc. for C_20_H_25_NO: C 81.31; H 8.53; N 4.74. Found: 81.30; H 8.55.; N 4.71. M = 295.19.

(±)-1-[Adamantan-1-yl(phenyl)methyl]amine hydrochloride (**6a**). 

To 1 g (3.75 mmol) of (±)-1-(isocyanato(phenyl)methyl)adamantane **5a** in 20 mL of anhydrous toluene, 0.5 mL of concentrated hydrochloric acid (4.1 mmol of HCl) was added with stirring and the reaction mass was kept for 1 h. The resulting white precipitate was filtered off, washed with acetonitrile, and dried, then recrystallized from water. The yield is 1.03 g (99%), white powder m.p. 262–263 °C. Mass spectrum, *m/z* (I rel. %): 241 (1% [M]^+^), 135 (12% [Ad]^+^), 106 (100% [Ph-CH_2_-NH_2_]^+^) (Appendix A). ^1^H NMR (300 MHz, DMSO-*d_6_*, δ) ppm: 1.25–1.75 m (14H, (Ad-CH(Ph)-NH_2_), 1.91 s (3H, Ad), 7.12–7.35 m (5H, arom) (Appendix A). Calc. for C_17_H_23_N: C 84.59; H 9.60; N 5.80. Found: 84.55; H 9.64.; N 5.75. M = 241.38.

Procedure for synthesis series of 1,3-disubstituted ureas **8a**–**j** and **10a**–**c**. Atom labels for ^13^C NMR presented on Figure 2.

(±)-1-((Adamantyl)(phenyl)methyl)-3-phenyl urea (**8a**).

**Method A.** To 200 mg (0.75 mmol) 1-(isocyanato(phenyl)methyl)-3,5-dimethyl -adamantane (**5a**) in 5 mL of diethyl ether, 80 mg (0.79 mmol) of triethylamine and 70 mg (0.75 mmol) of aniline (**7a**) were added. The reaction mixture was exposed at room temperature for 12 h. After adding 5 mL of 1 N HCl, the mixture was stirred for 1 h. The precipitated white residual matter was filtered and washed with water. The product was purified by recrystallization from ethanol. Yield of 189 mg (70%), m.p. 253–254 °C.

**Method B**. To 200 mg (0.72 mmol) hydrochloride 1-[adamantyl(phenyl)methyl] -amine (**6a**) in 5 mL of diethyl ether, 160 mg (1.58 mmol) of triethylamine and 70 mg (0.75 mmol) of phenylisocyanate (**9a**) were added. The reaction mixture was exposed at room temperature for 12 h. After adding 5 mL of 1 N HCl, the mixture was stirred for 1 h. The precipitated white residual matter was filtered and washed with water. The product was purified by recrystallization from ethanol. The yield is 221 mg (85%), m.p. 253–254 °C.

^1^H NMR (300 MHz, DMSO-*d_6_*, δ) ppm: 1.26–1.70 m (12H, Ad), 1.91 s (3H, Ad), 4.36 d (1H, Ad-CH(Ph)-, *J* = 9.2 Hz), 6.76–6.92 m (2H, Ad-CH(Ph)-NH-C(O)-NH-Ph-4H), 7.08–7.21 m (4H, Ph-NH), 7.24–7.40 m (5H, Ph-CH), 8.40 s (1H, ^3^NH) (Appendix A). ^13^C NMR (75 MHz, DMSO-*d_6_*, δ) ppm: 28.20 (**C^3^**, **C^5^**, **C^7^**), 36.32 (**C^1^**), 36.94 (**C^4^**, **C^6^**, **C^10^**), 38.91 (**C^2^**, **C^8^**, **C^9^**), 62.75 (**C^11^**), 117.73 (**C^20^**, **C^24^**), 121.38 (**C^15^**), 126.94 (**C^22^**), 127.82 (**C^14^**, **C^16^**), 128.71 (**C^21^**, **C^23^**), 129.12 (**C^13^**, **C^17^**), 140.81 (**C^19^**), 140.92 (**C^12^**), 155.17 (**C^18^**) (Appendix A). Calc. for C_24_H_28_N_2_O: C 79.96; H 7.83; N 7.77. Found: C 79.97; H 7.85; N 7.79. M = 360.22.

(±)-1-((Adamantan-1-yl)(phenyl)methyl)-3-(2-fluorophenyl) urea (**8b**). 

It was obtained similarly to the compound (**8a**), by **method A**, from 200 mg of the compound (**5a**) and 83 mg of 2-fluorophenine (**7b**). The yield is 71 mg (25%), m.p. 249–250 °C. It also was obtained by **method B**, from 200 mg of the compound (**6a**) and 98 mg of 2-fluorophenylisocyanate (**9b**). The yield is 218 mg (80%), m.p. 249–250 °C. ^1^H NMR (300 MHz, DMSO-*d_6_*, δ) ppm: 1.21–1.75 m (12H, Ad), 1.91 s (3H, Ad), 4.37 d (1H, Ad-CH(Ph), *J* = 8.9 Hz), 6.79–6.92 m (2H, NH-C(O)-NH-Ph-2F), 6.93–7.05 m (1H, NH-Ph-2F), 7.07–7.22 m (4H, Ad-CH(Ph)-NH-C(O)-NH-Ph-2F), 7.23–7.38 m (2H, Ph-CH), 8.01–8.20 m (1H, NH-Ph-2F), 8.40 s (1H, ^3^NH) (Appendix A). ^13^C NMR (75 MHz, DMSO-*d_6_*, δ) ppm: 28.17 (**C^3^**, **C^5^**, **C^7^**), 36.31 (**C^1^**), 36.90 (**C^4^**, **C^6^**, **C^10^**), 38.86 (**C^2^**, **C^8^**, **C^9^**), 62.96 (**C^11^**), 115.14 (d, *J* = 18.9 Hz) (**C^21^**), 119.96 (d, *J* = 2.0 Hz) (**C^23^**), 121.70 (d, *J* = 7.3 Hz) (**C^22^**), 124.81 (d, *J* = 3.4 Hz) (**C^24^**), 127.02 (**C^15^**), 127.87 (**C^14^**, **C^16^**), 128.69 (**C^13^**, **C^17^**), 128.893 (d, *J* = 10.1 Hz) (**C^19^**), 140.61 (**C^12^**), 151.77 (d, *J* = 239.9 Hz) (**C^20^**), 154.88 (**C^18^**) (Appendix A). ^19^F NMR (282 MHz, DMSO-*d_6_*, δ) ppm: -133.85 (1F) (Appendix A). Calc. for C_24_H_27_FN_2_O: C 76.16; H 7.19; N 7.40. Found: C 76.18; H 7.17; N 7.42. M = 378.21.

(±)-1-((Adamantan-1-yl)(phenyl)methyl)-3-(3-fluorophenyl) urea (**8c**).

It was obtained similarly to the compound (**8a**), by **method A**, from 200 mg of compound (**5a**) and 83 mg of 3-fluorophenine (**7c**). The yield is 83 mg (29%), m.p. 233–234 °C. ^1^H NMR (300 MHz, DMSO-*d_6_*, δ) ppm: 1.32–1.64 m (12H, Ad), 1.91 s (3H, Ad), 4.36 d (1H, Ad-CH(Ph), *J* = 9.3 Hz), 6.64 td (1H, ^1^NH, *J* = 8.4, 2.6 Hz), 6.93-7.00 m (2H, arom), 7.12–7.39 m (7H, Ad-CH(Ph)-NH-C(O)-NH-Ph-3F), 7.41 dt (1H, NH-Ph-3F, *J* = 12.3, 2.3 Hz), 8.71 s (1H, ^3^NH) (Appendix A). ^13^C NMR (75 MHz, DMSO-*d_6_*, δ) ppm: 28.19 (**C^3^**, **C^5^**, **C^7^**), 36.31 (**C^1^**), 36.92 (**C^4^**, **C^6^**, **C^10^**), 38.87 (**C^2^**, **C^8^**, **C^9^**), 62.81 (**C^11^**), 104.42 (d, *J* = 26.7 Hz) (**C^22^**), 107.62 (d, *J* = 21.3 Hz) (**C^20^**), 113.49 (**C^24^**), 126.99 (**C^15^**), 127.84 (**C^14^**, **C^16^**), 128.72 (**C^13^**, **C^17^**), 130.58 (d, *J* = 9.9 Hz) (**C^23^**), 140.63 (**C^12^**), 142.82 (d, *J* = 11.5 Hz) (**C^19^**), 154.99 (**C^18^**), 162.92 (d, *J* = 240.3 Hz) (**C^21^**) (Appendix A). ^19^F NMR (282 MHz, DMSO-*d_6_*, δ) ppm: -114.92 (1F) (Appendix A). Calc. for C_24_H_27_FN_2_O: C 76.16; H 7.19; N 7.40. Found: C 76.17; H 7.18; N 7.41. M = 378.21.

(±)-1-((Adamantan-1-yl)(phenyl)methyl)-3-(4-fluorophenyl) urea (**8d**).

It was obtained similarly to the compound (**8a**), by **method A**, from 200 mg of compound (**5a**) and 83 mg of 4-fluorophenine (**7d**). The yield is 153 mg (54%), m.p. 270–271 °C. ^1^H NMR (300 MHz, DMSO-*d_6_*, δ) ppm: 1.28–1.68 m (12H, Ad), 1.91 s (3H, Ad), 4.35 d (1H, Ad-CH(Ph), *J* = 9.2 Hz), 6.81 d (1H, ^1^NH, *J* = 9.4 Hz), 7.0 t (2H, NH-Ph-4F, *J* = 8.7 Hz), 7.10–7.44 m (7H, Ad-CH(Ph)-NH-C(O)-NH-Ph-F), 8.42 s (1H, ^3^NH) (Appendix A). ^13^C NMR (75 MHz, DMSO-*d_6_*, δ) ppm: 28.16 (**C^3^**, **C^5^**, **C^7^**), 36.29 (**C^1^**), 36.88 (**C^4^**, **C^6^**, **C^10^**), 38.87 (**C^2^**, **C^8^**, **C^9^**), 62.80 (**C^11^**), 115.59 (d, *J* = 22.1 Hz) (**C^22^**, **C^23^**), 119.39 (d, *J* = 7.6 Hz) (**C^20^**,**C^24^**), 127.01 (**C^15^**), 127.86 (**C^14^**, **C^16^**), 128.68 (**C^13^**, **C^17^**), 137.15 (d, *J* = 7.6 Hz) (**C^19^**), 140.67 (**C^12^**), 155.30 (**C^18^**), 157.29 (d, *J* = 237.0 Hz) (**C^22^**) (Appendix A). Calc. for C_24_H_27_FN_2_O: C 76.16; H 7.19; N 7.40. Found: C 76.16; H 7.20; N 7.42. M = 378.21.

(±)-1-((Adamantan-1-yl)(phenyl)methyl)-3-(3-chlorophenyl) urea (**8e**).

It was obtained similarly to the compound (**8a**), by **method A**, from 200 mg of compound (**5a**) and 96 mg of 3-chloraniline (**7e**). The yield is 77 mg (26%), m.p. 244–245 °C. ^1^H NMR (300 MHz, DMSO-*d_6_*, δ) ppm: 1.40–1.66 m (12H, Ad), 1.91 s (3H, Ad), 4.35 d (1H, Ad-CH(Ph), *J* = 9.3 Hz), 6.83–6.99 m (2H, Ad-CH(Ph)-NH-C(O)-NH-Ph-3Cl), 7.10–7.34 m (7H, Ad-CH(Ph)-NH-C(O)-NH-Ph-3Cl), 7.64 t (1H, NH-Ph-3Cl, *J* = 2.1 Hz), 8.63 s (1H, ^3^NH) (Appendix A). ^13^C NMR (75 MHz, DMSO-*d_6_*, δ) ppm: 27.72 (**C^3^**, **C^5^**, **C^7^**), 35.72 (**C^4^**, **C^6^**, **C^10^**), 36.46 (**C^1^**), 38.41 (**C^2^**, **C^8^**, **C^9^**), 62.39 (**C^11^**), 115.71 (**C^24^**), 116.66 (**C^20^**), 120.55 (**C^15^**), 126.57 (**C^22^**), 127.41 (**C^14^**, **C^16^**), 128.25 (**C^13^**, **C^17^**), 130.26 (**C^23^**), 133.19 (**C^21^**), 140.13 (**C^19^**), 141.95 (**C^12^**), 154.48 (**C^18^**) (Appendix A). Calc. for C_24_H_27_ClN_2_O: C 72.99; H 6.89; N 7.09. Found: C 72.98; H 6.92; N 7.11. M = 394.18.

(±)-1-((3,5-Dimethyladamantane-1-yl)(phenyl)methyl)-3-phenyl urea (**8f**). Atom labels for ^13^C NMR presented on Figure 3.

It was obtained similarly to the compound (**8a**), by **method A**, from 200 mg of compound (**5b**) and 63 mg of aniline (**7a**). The yield is 195 mg (74%), m.p. 69–70 °C. ^1^H NMR (300 MHz, DMSO-*d_6_*, δ) ppm: 0.74 d (6H, (CH_3_)_2_, *J* = 9.8 Hz), 1.00 dd (3H, Ad, *J* = 24.6, 12.8 Hz), 1.20 s (10H, Ad), 1.42 d (1H, Ad, *J* = 11.8 Hz), 1.98 d (1H, Ad, *J* = 12.6 Hz), 4.39 d (1H, Ad-CH(Ph), *J* = 9.3 Hz), 6.84 t (1H, ^1^NH, *J* = 7.6 Hz), 7.15 dd (3H, CH-Ph, *J* = 7.6, 4.4 Hz), 7.22 s (4H, arom), 7.15 dd (2H, NH-Ph, *J* = 11.2, 7.8 Hz), 8.36 s (1H, ^3^NH) (Appendix A). ^13^C NMR (75 MHz, DMSO-*d_6_*, δ) ppm: 29.16 (**C^1^**), 31.03 (**C^25^**, **C^26^**), 31.05 (**C^3^**), 31.09 (**C^5^**), 31.11 (**C^7^**), 38.14 (**C^2^**), 43.09 (**C^4^**, **C^10^**), 45.02 (**C^8^**), 45.22 (**C^9^**), 51.03 (**C^6^**), 62.23 (**C^11^**), 117.63 (**C^20^**, **C^24^**), 121.32 (**C^15^**), 126.90 (**C^22^**), 127.82 (**C^14^**, **C^16^**), 128.62 (**C^21^**, **C^23^**), 129.07 (**C^13^**, **C^17^**), 140.82 (**C^19^**), 140.89 (**C^12^**), 155.02 (**C^18^**) (Appendix A). Calc. for C_26_H_32_N_2_O: 80.37; H 8.30; N 7.21. Found: C 80.36; H 8.31; N 7.22. M = 388.56.

(±)-1-((3,5-Dimethyladamantane-1-yl)(phenyl)methyl)-3-(2-fluorophenyl) urea (**8g**).

It was obtained similarly to the compound (**8a**), by **method A**, from 200 mg of compound (**5b**) and 75 mg of 2-fluorophenine (**7b**). The yield is 80 mg (29%), m.p. 51–52 °C. ^1^H NMR (300 MHz, DMSO-*d_6_*, δ) ppm: 0.72 td (6H, (CH_3_)_2_, *J* = 8.9, 4.2 Hz), 0.89–1.32 m (13H, Ad), 4.26 td (1H, Ad-CH(Ph), *J* = 18.0, 9.8 Hz), 6.94–7.36 m (9H, arom), 8.39 s (1H, ^3^NH) (Appendix A). ^13^C NMR (75 MHz, DMSO-*d_6_*, δ) ppm: 29.22 (**C^1^**), 31.08 (**C^3^**), 31.13 (**C^25^**, **C^26^**), 31.15 (**C^5^**), 31.17 (**C^7^**), 38.41 (**C^2^**), 43.14 (**C^4^**, **C^10^**), 45.05 (**C^8^**), 45.22 (**C^9^**), 51.08 (**C^6^**), 63.14 (**C^11^**), 115.13 (d, *J* = 18.9 Hz) (**C^21^**), 119.96 (d, *J* = 2.0 Hz) (**C^23^**), 121.70 (d, *J* = 7.3 Hz) (**C^22^**), 124.81 (d, *J* = 3.4 Hz) (**C^24^**), 127.06 (**C^15^**), 127.80 (**C^14^**, **C^16^**), 127.81 (d, *J* = 10.1 Hz) (**C^19^**), 128.67 (**C^13^**, **C^17^**), 140.21 (**C^12^**), 151.77 (d, *J* = 239.9 Hz) (**C^20^**), 154.76 (**C^18^**) (Appendix A). Calc. for C_26_H_31_FN_2_O: C 76.81; H 7.69; N 6.89. Found: C 76.82; H 7.71; N 6.90. M = 406.24.

(±)-1-((3,5-Dimethyladamantane-1-yl)(phenyl)methyl)-3-(3-fluorophenyl) urea (**8h**).

It was obtained similarly to the compound (**8a**), by **method A**, from 200 mg of compound (**5b**) and 75 mg of 3-fluorophenine (**7c**). The yield is 85 mg (31%), m.p. 196–197 °C. ^1^H NMR (300 MHz, DMSO-*d_6_*, δ) ppm: 0.66–0.79 m (6H, (CH_3_)_2_), 0.84–1.32 m (13H, Ad), 4.21–4.44 m (1H, Ad-CH(Ph)), 6.93 t (1H, ^1^NH, *J* = 8.0 Hz), 6.97–7.34 m (7H, arom), 7.71 d (1H, Ph-F, *J* = 10.0 Hz), 8.59 s (1H, ^3^NH) (Appendix A). ^13^C NMR (75 MHz, DMSO-*d_6_*, δ) ppm: 29.22 (**C^1^**), 31.08 (**C^3^**), 31.12 (**C^25^**, **C^26^**), 31.15 (**C^5^**), 31.17 (**C^7^**), 38.40 (**C^2^**), 43.14 (**C^4^**, **C^10^**), 45.04 (**C^8^**), 45.25 (**C^9^**), 51.08 (**C^6^**), 63.13 (**C^11^**), 104.42 (d, *J* = 26.7 Hz) (**C^22^**), 107.62 (d, *J* = 21.3 Hz) (**C^20^**), 113.49 (**C^24^**), 127.04 (**C^15^**), 127.80 (**C^14^**, **C^16^**), 128.68 (**C^13^**, **C^17^**), 130.58 (d, *J* = 9.9 Hz) (**C^23^**), 140.76 (**C^12^**), 142.82 (d, *J* = 11.5 Hz) (**C^19^**), 154.88 (**C^18^**), 162.92 (d, *J* = 240.3 Hz) (**C^21^**) (Appendix A). Calc. for C_26_H_31_FN2O: C 76.81; H 7.69; N 6.89. Found: C 76.82; H 7.70; N 6.88. M = 406.24.

(±)-1-((3,5-Dimethyladamantane-1-yl)(phenyl)methyl)-3-(4-fluorophenyl) urea (**8i**).

It was obtained similarly to the compound (**8a**), by **method A**, from 200 mg of compound (**5b**) and 75 mg of 4-fluorine (**7d**). The yield is 168 mg (61%), m.p. 103–104 °C. ^1^H NMR (300 MHz, DMSO-*d_6_*, δ) ppm: 0.74 d (6H, (CH_3_)_2_, *J* = 9.1 Hz), 0.85–1.47 m (13H, Ad), 4.39 d (1H, Ad-CH(Ph), *J* = 9.4 Hz), 6.81 d (2H, Ph-F, *J* = 9.3 Hz), 7.00 t (1H, arom, *J* = 8.9 Hz), 7.14 d (2H, Ph-F, *J* = 7.6 Hz), 7.16–7.38 m (4H, arom), 8.39 s (1H, ^3^NH) (Appendix A). ^13^C NMR (75 MHz, DMSO-*d_6_*, δ) ppm: 29.23 (**C^1^**), 31.09 (**C^3^**), 31.12 (**C^25^**, **C^26^**), 31.16 (**C^5^**), 31.18 (**C^7^**), 38.20 (**C^2^**), 43.15 (**C^4^**, **C^10^**), 45.08 (**C^8^**), 45.27 (**C^9^**), 51.09 (**C^6^**), 62.32 (**C^11^**), 115.58 (d, *J* = 22.1 Hz) (**C^21^**, **C^23^**), 119.22 (d, *J* = 7.6 Hz) (**C^20^**, **C^24^**), 126.98 (**C^15^**), 127.90 (**C^14^**, **C^16^**), 128.68 (**C^13^**, **C^17^**), 137.25 (**C^19^**), 140.92 (**C^12^**), 155.14 (**C^18^**), 157.29 (d, *J* = 237.0 Hz) (**C^22^**) (Appendix A). Calc. for C_26_H_31_FN_2_O: C 76.81; H 7.69; N 6.89. Found: C 76.83; H 7.71; N 6.90. M = 406.24.

(±)-1-((3,5-Dimethyladamantane-1-yl)(phenyl)methyl)-3-(3-chlorophenyl) urea (**8j**).

It was obtained similarly to the compound (**8a**), by method A, from 200 mg of compound (**5b**) and 99 mg of 3-chloraniline (**7e**). The yield is 158 mg (53%), m.p. 111–112 °C. ^1^H NMR (300 MHz, DMSO-*d_6_*, δ) ppm: 0.73 t (6H, (CH_3_)_2_, *J* = 2.8 Hz), 0.76–1.46 m (13H, Ad), 4.38 d (1H, Ad-CH(Ph), *J* = 9.3 Hz), 6.83–6.97 m (1H, Ph-Cl), 6.99–7.35 m (7H, arom), 7.63 t (1H, Ph-Cl, *J* = 2.0 Hz), 8.58 s (1H, ^3^NH) (Appendix A). ^13^C NMR (75 MHz, DMSO-*d_6_*, δ) ppm: 29.18 (**C^1^**), 31.04 (**C^3^**), 31.08 (**C^25^**, **C^26^**), 31.11 (**C^5^**), 31.13 (**C^7^**), 38.14 (**C^2^**), 43.09 (**C^4^**, **C^10^**), 45.03 (**C^8^**), 45.23 (**C^9^**), 51.04 (**C^6^**), 62.41 (**C^11^**), 116.17 (**C^24^**), 117.11 (**C^20^**), 121.08 (**C^15^**), 127.07 (**C^22^**), 127.79 (**C^14^**, **C^16^**), 128.63 (**C^13^**, **C^17^**), 130.73 (**C^23^**), 133.62 (**C^21^**), 140.13 (**C^19^**), 142.26 (**C^12^**), 154.89 (**C^18^**) (Appendix A). Calc. for C_26_H_31_ClN_2_O: C 73.83; H 7.39; N 6.62. Found: C 73.85; H 7.41; N 6.63. M = 406.24.

(±)-(4-((4-(3-((3,5-Dimethyladamantan-1-yl)(phenyl)methyl)ureido)cyclohexyl)oxy) benzoic acid (**8k**). Atom labels for ^13^C NMR presented on Figure 4.

It was obtained similarly to the compound (**8a**), by method A, from 200 mg of compound (**5b**) and 160 mg of trans-4-(cyclohexyloxy)benzoic acid (**7f**). The yield is 184 mg (51%), m.p. 174–175 °C. ^1^H NMR (300 MHz, DMSO-*d_6_*, δ) ppm: 0.65–0.79 m (6H, (CH_3_)_2_), 0.87–1.03 m (2H, CH_2_ cyclohex), 0.98–1.24 m (13H, Ad), 1.25–1.53 m (2H, CH_2_ cyclohex), 4.05 s (1H, Ad-CH(Ph)), 4.15–4.43 m (1H, CH cyclohex), 6.98 m (2H, arom), 7.08–7.15 m (1H, arom), 7.16–7.35 m (4H, arom), 7.71 d (2H, arom, *J* = 10.0 Hz), 7.81–7.87 m (2H, NH-C(O)-NH), 8.00 br.s (1H, COOH) (Appendix A). ^13^C NMR (75 MHz, DMSO-*d_6_*, δ) ppm: 26.30 (**C^20^**), 29.23 (**C^1^**), 29.35 (**C^24^**), 31.05 (**C^3^**), 31.12 (**C^32^**, **C^33^**), 31.13 (**C^5^**), 31.17 (**C^7^**), 37.27 (**C^23^**), 38.14 (**C^21^**), 38.41 (**C^2^**), 43.14 (**C^4^**, **C^10^**), 45.03 (**C^8^**), 45.23 (**C^9^**), 51.08 (**C^6^**), 63.11 (**C^11^**), 64.63 (**C^19^**), 74.28 (**C^22^**), 115.60 (**C^26^**, **C^30^**), 120.32 (**C^28^**), 120.38 (**C^15^**), 127.80 (**C^14^**, **C^16^**), 128.68 (**C^13^**, **C^17^**), 131.84 (**C^27^**, **C^29^**), 140.21 (**C^12^**), 156.48 (**C^18^**), 161.35 (**C^25^**), 167.40 (**C^31^**) (Appendix A). Calc. for C_33_H_42_N_2_O_4_: C 74.69; H 7.98; N 5.28. Found, %: C 74.68; H 7.99; N 5.30. M = 530.71.

(±)-1-((Adamantan-1-yl)(phenyl)methyl)-3-(4-(trifluoromethoxy)phenyl) urea (**10a**). Atom labels for ^13^C NMR presented on Figure 5.

It was obtained similarly to the compound (**8a**) according to **method B** from 200 mg of compound (**5b**) and 146 mg of 4-(trifluoromethoxy)phenylisocyanate (**9c**). The yield is 181 mg (76%), m.p. 268–269 °C. ^1^H NMR (300 MHz, DMSO-*d_6_*, δ) ppm: 1.11–1.77 m (12H, Ad), 1.91 s (3H, Ad), 4.36 d (1H, Ad-CH(Ph), *J* = 8.5 Hz), 6.87 m (1H, ^1^NH), 7.10–7.49 m (9H, Ad-CH(Ph)-NH-C(O)-NH-Ph), 8.59 s (1H, ^3^NH) (Appendix A). ^13^C NMR (75 MHz, DMSO-*d_6_*, δ) ppm: 28.19 (**C^3^**, **C^5^**, **C^7^**), 36.33 (**C^1^**), 36.92 (**C^4^**, **C^6^**, **C^10^**), 38.888 (**C^2^**, **C^8^**, **C^9^**), 62.82 (**C^11^**), 118.82 (**C^21^**, **C^23^**), 120.67 (d, *J* = 253.5 Hz) (**C^25^**), 122.04 (**C^22^**, **C^24^**), 126.99 (**C^15^**), 127.84 (**C^14^**, **C^16^**), 128.69 (**C^13^**, **C^17^**), 140.19 (**C^19^**), 140.64 (**C^12^**), 142.45 (**C^22^**), 155.03 (**C^18^**) (Appendix A). ^19^F NMR (282 MHz, DMSO-*d_6_*, δ) ppm: -59.80 (3F) (Appendix A). Calc. for C_25_H_27_F_3_N_2_O_2_: C 67.55; H 6.12; N 12.82. Found, %: C 67.57; H 6.14; N 12.83. M = 444.20.

(±)-1-((Adamantan-1-yl)(phenyl)methyl)-3-(2-chlorophenyl) urea (**10b**).

It was obtained similarly to the compound (**8a**) by **method B** from 200 mg of the compound (**5b**) and 92 mg of 3-chlorophenylisocyanate (**9d**). The yield is 176 mg (83%), m.p. 265–266°C. ^1^H NMR (300 MHz, DMSO-*d_6_*, δ) ppm: 1.55 s (12H, Ad), 1.94 s (3H, Ad), 4.41 s (1H, Ad-CH(Ph)), 7.23 s (9H, arom), 8.16 s (2H, NH-C(O)-NH) (Appendix A). ^13^C NMR (75 MHz, DMSO-*d_6_*, δ) ppm: 28.18 (**C^3^**, **C^5^**, **C^7^**), 36.34 (**C^1^**), 36.91 (**C^4^**, **C^6^**, **C^10^**), 38.95 (**C^2^**, **C^8^**, **C^9^**), 63.13 (**C^11^**), 120.91 (**C^15^**), 121.22 (**C^20^**), 122.72 (**C^23^**), 122.82 (**C^24^**), 127.08 (**C^21^**), 127.90 (**C^14^**, **C^16^**), 128.78 (**C^13^**, **C^17^**), 129.54 (**C^22^**), 137.21 (**C^19^**), 140.58 (**C^12^**), 154.89 (**C^18^**) (Appendix A). Calc. for C_24_H_27_ClN_2_O: C 72.99; H 6.89; N 8.98. Found, %: C 72.98; H 6.88; N 8.99. M = 394.18.

(±)-1-((Adamantan-1-yl)(phenyl)methyl)-3-(cyclohexyl) urea (**10c**).

It was obtained similarly to the compound (**8a**) by **method B** from 200 mg of compound (**5b**) and 90 mg of cyclohexylisocyanate (**9e**). The yield is 156 mg (79%), m.p. 263–264 °C. ^1^H NMR (300 MHz, DMSO-*d_6_*, δ) ppm: 1.55 s (23H, Ad, 4-CH_2_), 1.89 es (2H, CH_2_), 4.29 s (1H, Ad-CH(Ph)-NH), 5.73 s (1H, ^1^NH), 6.37 s (1H, NH-C_6_H_11_), 7.23 s (5H, arom) (Appendix A). ^13^C NMR (75 MHz, DMSO-*d_6_*, δ) ppm: 24.79 (**C^21^**,**C^23^**), 25.77 (**C^22^**), 28.21 (**C^3^**, **C^5^**, **C^7^**), 33.76 (**C^20^**,**C^24^**), 36.42 (**C^1^**), 37.00 (**C^4^**, **C^6^**, **C^10^**), 38.91 (**C^2^**, **C^8^**, **C^9^**), 48.05 (**C^19^**), 62.65 (**C^11^**), 126.69 (**C^15^**), 127.67 (**C^14^**, **C^16^**), 128.74 (**C^13^**, **C^17^**), 141.37 (**C^12^**), 157.48 (**C^18^**) (Appendix A). Calc. for C_24_H_34_N_2_O: C 78.64; H 9.35; N 7.64. Found: C 78.66; H 9.36; N 7.65. M = 366.27.

## 4. Conclusions

The one-stage insertion of an adamantane moiety through the reaction of 1,3-dehydroadamantane and its 3,5-dimethyl homolog allowed us to obtain isocyanates containing a phenylmethylene fragment located between the adamantane fragment and the isocyanate group with a yield of 95% and 89%, respectively. The reaction of synthesized isocyanates with fluorine(chlorine)-containing anilines and trans-4-amino-(cyclohexyloxy)benzoic acid gave a series of 1,3-disubstituted ureas with 29–74% yield. The reaction of 1-[isocyanato-(phenyl)methyl]adamantane with fluoro(chlorine)-containing anilines gave a series of 1,3-disubstituted ureas with 25–85% yield. Inhibitory activity against sEH and other biochemical data for the synthesized compounds will be published in a further manuscript as soon as it can be acquired.

## Data Availability

Not applicable.

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
