# Peer review of "Synthesis and Properties of 1,3-Disubstituted Ureas Containing (Adamantan-1-yl)(phenyl)methyl Fragment Based on One-Pot Direct Adamantane Moiety Inclusion"

_molecules, 2023, doi:10.3390/molecules28083577_

Round 1

Reviewer 1 Report

In this manuscript, Burmistrov et al. describe the synthesis and properties of 1,3-disubstituted ureas containing 2-(adamantan-1-yl)(phenyl)methyl fragment based on one-pot direct adamantane moiety inclusion. Due to the importance of facile preparation of this kind of urea in organic chemistry, this direct synthetic route can be considered a practical way for the synthesis of these scaffolds. The final products are all new and were characterized by HNMR, mass spectrometry, and elemental analysis. Despite the value of the products, the manuscript has some vague and incomplete points that need to be improved for its acceptance. I believe these drawbacks must be corrected and elucidated in the manuscript.

1.       In previous years, Burmistrov and his colleagues have used similar synthetic methods for urea containing adamantyl, which they cited in the text of the article. Due to the similarity of this paper with previous works, the novelty of the manuscript is somewhat moderate. The authors should provide more reasons for the importance of the structure and the novelty of their work in the paper.

2.       Since the structures are new and the final product contains the carbonyl functional group, CNMR analysis needs to be conducted.

3.       In the spectral data part of the NMR, DMSO is used as the solvent, but the peaks related to the DMSO solvent are not seen in the supporting information file, such as compounds 8a-c.

4.       This manuscript needs to have a clear graphical abstract.

5.       In order to help readers better understand, it is better to show the two reaction components in the product with different colors. It may be difficult for readers to follow when using adamantyl amines or adamantyl isocyanate.

6.       In the abstract, the author claims that "The resulting 1,3-disubstituted ureas are promising inhibitors of the human soluble epoxide hydrolase (hsEH) and the high lipophilicity of adamantane allows it to pass through the blood-brain barrier." However, there is no solid evidence in the manuscript to support this claim. I highly recommend the author conduct some computational studies, such as molecular docking or molecular dynamic investigations.

7.       Diethyl ether is chosen as the solvent, and triethylamine is used as the base for this reaction, but it is better to examine this reaction first in greener and more stable solvents, such as water, ethanol, isopropanol, ethyl acetate, and so on. Moreover, other bases, such as inorganic bases, should be examined in this reaction.

8.       In the introduction part, the sentence “A research group from the Institute of Biomedicine of the University of Barcelona (IBUB), led by Professor Santiago Vazquez, considered the replacement of the adamantane fragment with compounds containing polycyclic hydrocarbons smaller or larger than adamantane.” is better to replace with “Vazquez  et al considered the replacement of the adamantane fragment with compounds containing polycyclic hydrocarbons smaller or larger than adamantane.”

Author Response

In this manuscript, Burmistrov et al. describe the synthesis and properties of 1,3-disubstituted ureas containing 2-(adamantan-1-yl)(phenyl)methyl fragment based on one-pot direct adamantane moiety inclusion. Due to the importance of facile preparation of this kind of urea in organic chemistry, this direct synthetic route can be considered a practical way for the synthesis of these scaffolds. The final products are all new and were characterized by HNMR, mass spectrometry, and elemental analysis. Despite the value of the products, the manuscript has some vague and incomplete points that need to be improved for its acceptance. I believe these drawbacks must be corrected and elucidated in the manuscript.

  1. In previous years, Burmistrov and his colleagues have used similar synthetic methods for urea containing adamantyl, which they cited in the text of the article. Due to the similarity of this paper with previous works, the novelty of the manuscript is somewhat moderate. The authors should provide more reasons for the importance of the structure and the novelty of their work in the paper.

Compounds containing bulky lipophilic fragments can be used to study differences between soluble epoxide hydrolases of different species due to the differences the protein structures. Sentence was added to the introductory part of the manuscript.

  1. Since the structures are new and the final product contains the carbonyl functional group, CNMR analysis needs to be conducted.

13C and 19F NMR data added to the manuscript and supporting information file.

  1. In the spectral data part of the NMR, DMSO is used as the solvent, but the peaks related to the DMSO solvent are not seen in the supporting information file, such as compounds 8a-c.

Correct spectral data uploaded to the supporting information file.

  1. This manuscript needs to have a clear graphical abstract.

New graphical abstract prepared for the manuscript.

  1. In order to help readers better understand, it is better to show the two reaction components in the product with different colors. It may be difficult for readers to follow when using adamantyl amines or adamantyl isocyanate.

Colors for amine (Blue) and isocyanate (Red) groups added to Schemes 2,3,4

  1. In the abstract, the author claims that "The resulting 1,3-disubstituted ureas are promising inhibitors of the human soluble epoxide hydrolase (hsEH) and the high lipophilicity of adamantane allows it to pass through the blood-brain barrier." However, there is no solid evidence in the manuscript to support this claim. I highly recommend the author conduct some computational studies, such as molecular docking or molecular dynamic investigations.

Line about “passing through the blood-brain barrier” was removed from the abstract as it has no strong evidence and was based on previous studies of similar compounds.

Inhibitory activity and other biochemical data will be published as soon as it will be acquired in further manuscript for IJMS. However inhibitory activity of very similar compounds and compounds with even bigger lipophilic groups has no doubt and corresponding publication cited in Introduction.

  1. Diethyl ether is chosen as the solvent, and triethylamine is used as the base for this reaction, but it is better to examine this reaction first in greener and more stable solvents, such as water, ethanol, isopropanol, ethyl acetate, and so on. Moreover, other bases, such as inorganic bases, should be examined in this reaction.

Reviewer raises a good point. However, diethyl ether and triethylamine are gold standard for urea synthesis. Ureas usually insoluble in ether while most of amines and isocyanates as well as triethylamine are soluble. For the cases when starting material is insoluble in ether DMF is used. As for the water and alcohols they cannot be used as solvents for this reaction because will react with isocyanates. Most of other polar solvents such as ethyl acetate can dissolve resulting ureas which makes isolation more difficult. Inorganic bases also insoluble in most of solvents suitable for this reaction.

Moreover, right now we are studying some green solvents such as dihydrolevoglucosenone (Cyrene) for urea synthesis. This data will be presented in our next manuscript.

  1. In the introduction part, the sentence “A research group from the Institute of Biomedicine of the University of Barcelona (IBUB), led by Professor Santiago Vazquez, considered the replacement of the adamantane fragment with compounds containing polycyclic hydrocarbons smaller or larger than adamantane.” is better to replace with “Vazquez  et al considered the replacement of the adamantane fragment with compounds containing polycyclic hydrocarbons smaller or larger than adamantane.”

Corrected

Thank you for the review.

Reviewer 2 Report

The article of Burmistrov and co-workers describes the synthesis and some properties of ureas containing an adamantyl fragment.

They propose a short sequence to access the isocyanate precursor which is then coupled with aliphatic or aromatic amines to provide the desired ureas in moderate to high yields.

A “discussion” on the calculated lipophilicity coefficient LogP is then provided as long as a similar discussion based on the melting temperatures of the prepared ureas.

Overall, this study describes a short access to a specific class of ureas containing a sterically hindered adamantyl fragment that may be useful, from a practical point of view, for biological applications. This paper may thus be accepted in Molecules after some modifications.

Page 3 : Method A and B should be clearly expressed in Scheme 3 and 4. The difference is only visible in the experimental part

Page 6: discussion on the lipophilicity coefficient and melting point is not really a discussion as while differences in LoP and mp are observable from one compound to another, there is no real attempt at rationalizing these differences. What conclusions can be drawn from these differences depending on the nature of the substituents ?   

Author Response

The article of Burmistrov and co-workers describes the synthesis and some properties of ureas containing an adamantyl fragment. They propose a short sequence to access the isocyanate precursor which is then coupled with aliphatic or aromatic amines to provide the desired ureas in moderate to high yields. A “discussion” on the calculated lipophilicity coefficient LogP is then provided as long as a similar discussion based on the melting temperatures of the prepared ureas. Overall, this study describes a short access to a specific class of ureas containing a sterically hindered adamantyl fragment that may be useful, from a practical point of view, for biological applications. This paper may thus be accepted in Molecules after some modifications.

Page 3 : Method A and B should be clearly expressed in Scheme 3 and 4. The difference is only visible in the experimental part

Colors for amine (Blue) and isocyanate (Red) groups added to Schemes 2,3,4 along with reaction conditions.

Page 6: discussion on the lipophilicity coefficient and melting point is not really a discussion as while differences in LoP and mp are observable from one compound to another, there is no real attempt at rationalizing these differences. What conclusions can be drawn from these differences depending on the nature of the substituents ?  

Rationalization for the lipophilicity coefficient and melting points of the synthesized compounds added to the manuscript.

Thank you for the review.

Reviewer 3 Report

This paper describes the strategy route to access 1,3-Disubstituted Ureas containing (Adamantan-1-yl)(phenyl)methyl Fragment based on one-pot direct adamantane moiety inclusion. Also, the authors describe the properties of obtained compounds. The utility of these new intermediates was demonstrated. The method has the potential to be used to get more complex products despite the low chemical yields reported. All the characterization was done properly, and the Supporting Information is well described. Although the work does not advance significantly in the field it has its synthetic utility and deserves to be shared with the organic synthesis community. For all these reasons, I think this paper should be accepted for publication in this Journal.

Author Response

This paper describes the strategy route to access 1,3-Disubstituted Ureas containing (Adamantan-1-yl)(phenyl)methyl Fragment based on one-pot direct adamantane moiety inclusion. Also, the authors describe the properties of obtained compounds. The utility of these new intermediates was demonstrated. The method has the potential to be used to get more complex products despite the low chemical yields reported. All the characterization was done properly, and the Supporting Information is well described. Although the work does not advance significantly in the field it has its synthetic utility and deserves to be shared with the organic synthesis community. For all these reasons, I think this paper should be accepted for publication in this Journal.

Thank you for the review.

Reviewer 4 Report

In this manuscript, Burmistrov and co-workers described an one-stage method for the preparation of 1-[isocyanato(phenyl)methyl]adamantane containing a phenylmethylene fragment located between the adamantane fragment and the isocyanate group, and 1-[isocyanato(phenyl)methyl]-3,5-dimethyladamantane with additional methyl groups at the nodal positions of adamantane, with a yield of 95% and 89% respectively. Although several 1,3-disubstituted ureas containing (adamantan-1-yl)(phenyl) methyl fragments were successfully synthesized in this paper. However, the reviewer did not find any innovations that could be published on Molecules. In the introduction section, the authors highlight the potential of these obtained molecules for lipophilic applications, but there are no data available in this manuscript. Furthermore, the manuscript is full of clerical errors, so I do not recommend that it be accepted for publication in its current form.

Some of the errors are as follows:

1)      Table 1, the “Mr” should be “Mz”. “360,22” should be correct as “360.22”, “mp” should be correct as “m.p.”

2)      The NMR data of the compound should indicate the frequency.

3)      The authors should label the products in Table 1 with 1NH, 3NH, etc.

Author Response

In this manuscript, Burmistrov and co-workers described an one-stage method for the preparation of 1-[isocyanato(phenyl)methyl]adamantane containing a phenylmethylene fragment located between the adamantane fragment and the isocyanate group, and 1-[isocyanato(phenyl)methyl]-3,5-dimethyladamantane with additional methyl groups at the nodal positions of adamantane, with a yield of 95% and 89% respectively. Although several 1,3-disubstituted ureas containing (adamantan-1-yl)(phenyl) methyl fragments were successfully synthesized in this paper. However, the reviewer did not find any innovations that could be published on Molecules. In the introduction section, the authors highlight the potential of these obtained molecules for lipophilic applications, but there are no data available in this manuscript. Furthermore, the manuscript is full of clerical errors, so I do not recommend that it be accepted for publication in its current form.

Some of the errors are as follows:

1)      Table 1, the “Mr” should be “Mz”. “360,22” should be correct as “360.22”, “mp” should be correct as “m.p.”

Corrected. Also corrected in Experimental section

2)      The NMR data of the compound should indicate the frequency.

Frequency added

3)      The authors should label the products in Table 1 with 1NH, 3NH, etc.

Labeling atoms in the Table 1 seems to be overloading. We added structures with labelled N and C atom to the experimental section which can make spectral data easier to understand.

Thank you for the review. In this manuscript we primarily wanted to report a one-stage method for the adamantane inclusion which opens simple way for the preparation of isocyanates and ureas. Inhibitory activity and other biochemical data will be published as soon as it will be acquired in further manuscript for IJMS.

Round 2

Reviewer 1 Report

The questions and requests have been well answered by the authors of this article.

A few points still remain, however.

First, the final products in Table 1 must be coloured and determined which part is isocyanide and amine.

Second, the authors assert that "Inhibitory activity and other biochemical data will be published as soon as it will be acquired in further manuscript". they need to pinpoint that at the end of the conclusion part.

Third, the authors answered my question regarding solvent and base effects in this reaction, I highly recommended that it is added to the result and discussion part.

Author Response

The questions and requests have been well answered by the authors of this article.

A few points still remain, however.

First, the final products in Table 1 must be coloured and determined which part is isocyanide and amine.

Compounds in Table 1 were coloured.

Second, the authors assert that "Inhibitory activity and other biochemical data will be published as soon as it will be acquired in further manuscript". they need to pinpoint that at the end of the conclusion part.

Statement added to the conclusion part.

Third, the authors answered my question regarding solvent and base effects in this reaction, I highly recommended that it is added to the result and discussion part.

Solvent discussion added.

Thank you for the review!

Reviewer 4 Report

The author addressed my concerns.

Author Response

Thank you for the review!